# Green Synthesis of Zinc Oxide (ZnO) Nanoparticles from Green Algae and Their Assessment in Various Biological Applications

**DOI:** 10.3390/mi14050928

**Published:** 2023-04-25

**Authors:** Hajra Hameed, Abdul Waheed, Muhammad Shakeeb Sharif, Muhammad Saleem, Afshan Afreen, Muhammad Tariq, Asif Kamal, Wedad A. Al-onazi, Dunia A. Al Farraj, Shabir Ahmad, Rania M. Mahmoud

**Affiliations:** 1Department of Biotechnology, Mirpur University of Science and Technology, New Mirpur City 10250, Pakistan; hajrahameedmughal@gmail.com (H.H.); mshakeebsharif@gmail.com (M.S.S.); muhammadsaleem786687@gmail.com (M.S.); 2Agricultural Genomics Institute at Shenzhen, Chinese Academy of Agricultural Sciences, Shenzhen 518120, China; 2021y90100044@caas.cn; 3Department of Plant Sciences, Quaid-i-Azam University, Islamabad 45320, Pakistan; a.kamal@bs.qau.edu.pk; 4Department of Chemistry, College of Science, King Saud University, Riyadh 11495, Saudi Arabia; walonazi@ksu.edu.s; 5Department of Botany and Microbiology, College of Science, King Saud University, Riyadh 11495, Saudi Arabia; dfarraj@ksu.edu.sa; 6Department of Botany and Biodiversity Research, University of Vienna, 1010 Vienna, Austria; ahmads58@univie.ac.at; 7Department of Botany, Faculty of Science, University of Fayoum, Fayoum 63514, Egypt; rmm00@fayoum.edu.eg

**Keywords:** zinc oxide, nanoparticles, *Spirogyra hyalina*, antibacterial, antioxidant

## Abstract

The biosynthesis of algal-based zinc oxide (ZnO) nanoparticles has shown several advantages over traditional physico-chemical methods, such as lower cost, less toxicity, and greater sustainability. In the current study, bioactive molecules present in *Spirogyra hyalina* extract were exploited for the biofabrication and capping of ZnO NPs, using zinc acetate dihydrate and zinc nitrate hexahydrate as precursors. The newly biosynthesized ZnO NPs were characterized for structural and optical changes through UV-Vis spectroscopy, Fourier transform infrared spectroscopy (FT-IR), X-ray diffraction (XRD), scanning electron microscopy (SEM), and energy dispersive X-ray spectroscopy (EDX). A color change in the reaction mixture from light yellow to white indicated the successful biofabrication of ZnO NPs. The UV-Vis absorption spectrum peaks at 358 nm (from zinc acetate) and 363 nm (from zinc nitrate) of ZnO NPs confirmed that optical changes were caused by a blue shift near the band edges. The extremely crystalline and hexagonal Wurtzite structure of ZnO NPs was confirmed by XRD. The involvement of bioactive metabolites from algae in the bioreduction and capping of NPs was demonstrated by FTIR investigation. The SEM results revealed spherical-shaped ZnO NPs. In addition to this, the antibacterial and antioxidant activity of the ZnO NPs was investigated. ZnO NPs showed remarkable antibacterial efficacy against both Gram-positive and Gram-negative bacteria. The DPPH test revealed the strong antioxidant activity of ZnO NPs.

## 1. Introduction

Phytonanotechnology refers to the biogenic synthesis of nanomaterials by exploiting the bioactive molecules, including polyphenols, alkaloids, proteins, etc., of biological entities, such as plants, bacteria, and algae. The phytochemicals act as bioreducing, stabilizing, and capping agents [1]. Phytochemicals from plants and algae, as well as bioactives from other natural sources, are frequently used for the biosynthesis of metal and metal oxide nanoparticles. This innovative approach has many different applications in the fields of agriculture, medicine, biotechnology, and the food industry. The nanoparticles prepared by conventional physical and chemical methods are less effective clinically, owing to their toxicity [2,3]. Moreover, conventional physico-chemical procedures use hazardous raw materials, are conducted under harsh conditions, including temperature, pressure, and energy, and produce toxic by-products [4,5]. Therefore, interest among researchers in phytonanotechnology is growing rapidly. The green synthesis of NPs is environmentally friendly, cost-effective, easy, upscale-able, and biocompatible [6]. These biosynthesized NPs are used as antimicrobial, anticancer, and drug delivery agents. Biological entities such as plants, fungi, algae, etc., have a repertoire of bioactive molecules such as polyphenols, flavanols, carbohydrates, proteins, vitamins, and coenzyme-based intermediates. These phytochemicals contain many organic functional groups, such as hydroxyl, carboxylic, amine, and carbonyl, that are involved in the bioreduction of metal ions in the nano range in a single reaction at room temperature [7,8]. These biologically active molecules act as capping agents for NPs, which is essential for their stability and biocompatibility [9]. 

The potential use of metal and metal oxide NPs depends upon the metal used for the biogenic synthesis of NPs. Several metal and metal oxide NPs were fabricated using green methods, including Au, Ag, Se, Cu, CuO, Fe_3_O_4_, ZnO, and many others, for various biomedical applications [1,10,11]. These biogenic metal and metal oxide NPs are highly toxic to animals and humans, which has hampered their use for medical purposes. Zinc oxide (ZnO) is a rare natural inorganic crystalline compound that is found in a red or orange color due to manganese impurities [12], while purified zinc oxide is a white crystalline powder that is almost insoluble in water. ZnO NPs are widely utilized in the textile, cosmetic, and micro-electronic industries owing to their low toxicity and size-dependent properties [2]. ZnO is generally recognized as safe (GRAS) and has antimicrobial properties; this is why ZnO NPs are promising agents for the treatment of infections in humans and animals [13]. ZnO NPs are also employed for diagnostic purposes [1]. 

*Spirogyra* is a genus of freshwater filamentous green algae of the family Zygnemataceae, with over 400 species in the world [14]. The rich nutritional components of *Spirogyra* sp., including its proteins, fats, carbohydrates, minerals, vitamins, and antioxidants, make it a popular food in Thailand [15]. *Spirogyra* spp. were reported to have antibacterial and antifungal activities [14]. The phytochemicals obtained from macroalgae are commonly used for cosmetic and pharmaceutical purposes. Another study reported the presence of phytochemicals such as alkaloids, phenols, flavonoids, steroids, tannins, and terpenoids in *Spirogyra* and their antimicrobial potential against two bacterial strains, *Escherichia coli* and *Candida albicans* [15]. Many researchers harnessed algae for the biofabrication of metallics, such as Ag, Au, Cu, and Se, and metallic oxide nanoparticles, such as Fe_3_O_4_, CuO, and ZnO, such as those from *Calothrix* sp., *Corallina elongate*, *Spirogyra hyalina*, *Spirogyra varians*, *Gelidium crinale*, *Sargassum muticum*, *Turbinaria turbinate*, *Laurencia obtuse*, *Laurencia aldingensis*, *Laurenciella* sp., *Bifurcaria bifurcate*, *Ulva fasciata*, and *Cystoseira myrica* [15,16,17,18,19,20]. 

In this study, ZnO NPs were biofabricated using a simple and eco-friendly technique using the extract of the green alga *Spirogyra hyalina* as a reducing agent and zinc acetate and zinc nitrate as precursors. The biosynthesized ZnO NPs were characterized with UV–Vis spectroscopy, FTIR, XRD, SEM, and EDX. Furthermore, bioactivities, such as antioxidant and antimicrobial potential against pathogenic Gram-positive and Gram-negative bacteria, were assessed and compared with a crude extract of algae. It is expected that the present research will enhance the usage of algae-based ZnO nanoparticles in the biomedical and cosmetic industries.

## 2. Results

### 2.1. UV-Visible Spectroscopy

ZnO NP formation was examined by the color changes of solution mixtures from light yellow to a light apple greenish color of ZnO NPs (ZA) and a pure white color in the case of ZnO NPs (ZN). The synthesis of ZnO NPs was further investigated by UV-vis spectroscopy.

ZnO nanoparticles prepared from green algae have shown sharp absorbance peaks at 358 nm and 363 nm of zinc acetate dihydrate precursor and zinc nitrate hexahydrate precursor, respectively, as shown in Figure 1. These results satisfy the standard ZnO absorption pattern because all oxide materials have wide band gaps and tend to have shorter wavelengths. Moreover, if the material is in nanoscale, it tends to has even shorter wavelengths. This notion supports the results observed for ZnO NPs here.

### 2.2. Fourier-Transform Infrared Spectroscopy Analysis

The functional group involvement in the biological compounds of algal extract in the bioreduction of ZnO NPs was detected by FTIR analysis. The border peaks of algal extract at 3300–3400 cm^−1^ corresponded to the OH-hydroxyl group of alcohols or phenolic compounds. The absorbance peaks observed at 3000 and 3050 cm^−1^ referred to the CH bond of an alkane’s functional group in the stretching position of the spectra. The band at 1600–1550 cm^−1^ corresponded to the amide I–II region due to the carbonyl stretching of proteins. The peaks from 1000 to 950 cm^−1^ corresponded to C-N and C-H bending due to phytochemicals, such as aromatic and aliphatic amines, in algal extract. The FTIR absorption spectra in Figure 2 revealed a sharp absorbance peak at 500 cm^−1^, which is the characteristic Zn-O peak, which affirmed the biosynthesis of ZnO NPs via phytochemical synthesis. Although few changes were observed at these frequencies, most peaks in the FT-IR spectrum showed a shift towards lower frequencies and a decrease in the intensity of binding with the nanoparticles. This trend of free carbonyl and NH_2_ groups from proteins and amino acid residues indicates that they have the ability to bind to a metal and that the proteins could possibly form a layer around the metal for preventing agglomeration and thereby stabilizing the nanoparticles. These findings suggest that not only the OH group of flavonoids but also the protein molecules and their functional groups play an important role in the bioreduction of salts and the capping of NPs.

### 2.3. XRD Analysis

An X-ray diffractometer used a powdered sample of zinc oxide nanoparticles in order to analyze the zinc oxide nanoparticles structurally. The XRD pattern of biosynthesized zinc oxide nanoparticles showed their highly crystalline structure. As shown in Figure 3, the bio-fabricated nanoparticles depicted seven distinct peaks corresponding to 2 theta degrees: 31.8°, 34.5°, 36.3°, 47.6°, 56.7°, 63.0°, and 66.5°, which matched the (100), (002), (101), (102), (110), (103), and (200) planes, respectively.

The XRD pattern of both ZnO NPs (ZA) and ZnO NPs (ZN) corresponded to the planes of ZnO in the Wurtzite Structure, which corresponded to JCPDS Card No. 01-075-0576. The average crystal size was calculated from the XRD analysis using the Debye–Scherrer equation, which, in this study, was approximately equal to ≈45 nm. The data from XRD were compatible with those obtained by SEM analysis. The presence of slight peaks in the XRD spectra may be related to the crystallization of organic substances that coated the surface of ZnO NPs.

### 2.4. SEM Images of ZnO NPS

SEM analysis revealed that biofabricated ZnO NPs (ZA) and (ZN) were spherical in shape, with some aggregation seeming to be rough, as shown in Figure 4. The SEM results of ZnO NPs (ZA) showed size ranges from 50 nm to 80 nm, with an average size of 65 nm, and ZnO (ZN) size ranges from 20 nm to 60 nm, with an average size of 40 nm. The size of NPs were evaluated with the help of Image J and Origin softwares. The shape of NPs plays a very crucial role in their effectiveness against pathogens, because spherical NPs tend to be very potent during antibacterial activity owing to their ability to easily penetrate the cell wall of pathogens. ZnO NPs synthesized from algae can be of great importance in treating clinical pathogens.

### 2.5. EDX Analysis

The quantitative elemental structure of greenly synthesized ZnO NPs was assessed by EDX analysis, as shown in Figure 5 and Figure 6. It was clearly revealed from the “EDX spectrum” of ZnO NPs using zinc acetate and zinc nitrate as salt precursors, that there are two major peaks: zinc at 1 keV and oxygen at 0.5 keV, respectively, which indicate the biogenic fabrication of ZnO NPs. The EDX of ZnO NPs of zinc acetate and zinc nitrate salt precursors determined the elemental formulation and the mass percents of zinc and oxygen, which were 72.95%, 12.16%, 54.13%, and 30.54%, respectively. The EDX graph also showed a carbon peak at 0.3 keV with a mass of 14.89% and 15.33%, respectively. It might be appeared due to the presence of various phytochemicals present in algal extract.

### 2.6. Antibacterial Activity

ZnO NPs (ZA) and (ZN) precursors exhibited varied activities against *Pseudomonas aeruginosa*, *Bacillus pumilus*, *Staphylococcus aureus*, and *Escherichia coli*, with clear zones of 15.33 ± 0.42, 16.33 ± 1.14, 24.33 ± 1.12, and 16.67 ± 0.92 mm, respectively. It was clear from the antibacterial plates and graphs in Figure 7, Figure 8, Figure 9 and Figure 10 that by increasing the concentration of nanoparticles, the zones of inhibition against bacterial strains also increased. Zinc oxide NPs had strong antibacterial activity against Gram-positive bacteria, particularly *S. aureus*. Overall, it was evident from the antibacterial graphs that algae-based ZnO NPs showed high antibacterial activity compared to that of algae extract against the various types of Gram-positive and Gram-negative bacterial strains.

### 2.7. Antioxidant Activity

The antioxidant activity of synthesized ZnO NPs was determined by the DPPH assay shown in Figure 11 and Figure 12. The algal-based ZnO NPs showed remarkable scavenging potential (IC_50_ = 16.1 ± 0.74 µg/mL and 18.89 ± 0.63 µg/mL) compared to that of algal extract (IC_50_ values of 21.2 ± 0.98 µg/mL). A low IC_50_ value of a substance means that it has higher scavenging activity. The graphs revealed that algal-based ZnO NPs have lower IC_50_ values, which show the higher antioxidant potential. This might be due to the increased activity of phenolic molecules after the fabrication of ZnO NPs. Phenolic active agents found in algae extract play a significant role in producing ZnO nanoparticles due to their high antioxidant activity and ability to act as reducing agents.

## 3. Discussion

Nanotechnology is a developing field that serves as the exceptional foundation of multidisciplinary fields, coordinating a diverse range of fields such as biology, chemistry, and physics. In the present study, ZnO NPs were synthesized from an extract of *Spirogyra hyalina* using zinc acetate and zinc nitrate as precursor salts, their synthesis was initially confirmed by UV-visible spectroscopy. Further characterization of ZnO NPs was performed by FTIR, XRD, SEM and EDS, which confirmed the size and shape of ZnO nanoparticles at the nanoscale. Talam et al. (2012) and Srinivasan synthesized zinc oxide nanoparticles from zinc nitrate, and studied their optical properties, the nanoparticles showed an absorption peak at 355 nm [21,22]. ZnO nanoparticles were synthesized by means of a phytochemical route through a zinc acetate dihydrate precursor, *Calotropis* leaf extraction, and tea leaves showed the absorption peaks close to our findings [23,24,25]. Another recent study reported the green fabrication of ZnO nanoparticles using the leaf extract of *Cayratia pedate* also supported our results [26]. Sharmila et al. (2018) biosynthesized zinc oxide nanoparticles from the leaf extract of *Bauhinia tomentosa*, and their FTIR analysis are consisted with our results. Their zinc oxide nanoparticles additionally showed an absorption band at 3300 cm^−1^, which indicated OH group vibrations. The distinguished C-H bond was in stretching mode from 2800 to 2900 cm^−1^, similar to our results [27]. Zinc oxide nanoparticles prepared from the *Artocarpus gomezianus* natural product showed comparable FTIR peaks, as revealed by our study [28]. The FTIR spectra of *Murayya koenigii* leaf extract, which is related to *Murraya paniculata*, showed that the absorbance peak from 1500 to 1600 cm^−1^ showed -C=C- stretching and amide I and II regions due to carbonyl stretching in proteins, which is consistent with our findings [29]. In recent publications, it was indicated that ZnO nanoparticles produced by using *Sambucus ebulus* and *Calotropis gigantea*, which showed O-H bond stretching at 3429 cm^−1^, C-H bonds at 2900 cm^−1^, and ZnO presence at 545 cm^−1^, are related to our study [29,30]. A review described that the absorbance peak at 900 cm^−1^ alludes to Zn-OH, while Zn-O shows a quality peak at 500 cm^−1^, which is similar to our result [31]. In a recently reported study involved a single-step synthesis of zinc oxide nanoparticles from the extract of *Moringa oleifera*, showed the diffraction peaks with comparative planes, affirming the hexagonal Wurtzite structure of ZnO NPs in XRD also supported our findings [32]. ZnO NPs produced using zinc acetate and zinc nitrate salt precursors showed spherical-shaped nanoparticles [23], sizes ranging from 30 to 60 nm [32,33], and *Azadirachta indica* seed husk extraction [34], with a size range from 25 nm to 60 nm. Other studies found that ZnO NPs synthesized from *Trifolium pretense* flower extract have a diameter of 60–70 nm [35], ZnO NPs synthesized from *Sechium edule* leaf extract have a diameter of 30–70 nm [36], and ZnO NPs synthesized from *Bauhinia* reported the green fabrication of ZnO nanoparticles using seaweed via green chemistry [37], all of these SEM findings are consistent with our results. The EDX analysis of the respective zinc oxide nanoparticles showed higher mass percents of zinc at 1 eV and oxygen at 0.5 eV is 78.32%, 12.78%, and 52%, 48% respectively which are nearly close to those of currently synthesized zinc oxide nanoparticles [36,37]. In 2020, a recently published paper described the detailed green synthesis of zinc oxide nanoparticles by the leaf extract of *Koserat* and its fantastic antibacterial properties against Gram-positive and Gram-negative microbes [38]. The biosynthesis of ZnO nanoparticles from the leaf extract of *Bauhinia tomentosa* previously revealed that, ZnO nanoparticles showed higher antibacterial activity against Gram-positive and Gram-negative microscopic organisms [27]. A review indicated that the green chemistry of zinc oxide nanoparticles from *Coptidis rhizoma* and the scavenging potential of phytochemically synthesized ZnO nanoparticles with various concentrations showed that metal zinc oxide nanoparticles have effective antioxidant potential [39,40,41].

## 4. Materials and Methods

### 4.1. Sampling, Identification, and Extraction of Algae

Algae samples were collected from the Jatlan Head of Mirpur Azad Kashmir, Pakistan. The morphology of collected samples was examined under a light microscope to check the shape, color, and structure of *Spirogyra*. Microscopic taxonomy was assessed according to Prescott [42]. The algae were identified as *Spirogyra hyalina*. The algae were washed with distilled water and left to air dry. After air drying, the sample was soaked in methanol for one week and placed in a shaker. After one week, the filtrate was concentrated through a rotatory evaporator. The semisolid algae extract was stored in the fridge until further use.

### 4.2. Biosynthesis of ZnO Nanoparticles

For the biosynthesis of ZnO nanoparticles, 5 g of crude methanolic algal extract was mixed with 500 mL of distilled water, heated to 70 °C for 30 min, and filtered with muslin cloth and Whatman filter paper no. 1, respectively. The resultant pale yellowish-green algal extract (20 mL) was mixed with 80 mL of 0.02 M precursor salt (Zinc acetate or Zinc nitrate). The reaction mixture was stirred for 2 h. The NPs, in the form of white precipitate, started to settle at the bottom of the flask. The supernatant was discarded, and precipitates were transferred to 1.5 mL Eppendorf tubes. Both samples were washed three times with distilled water and centrifuged at 6000 rpm for 15 min.

### 4.3. Characterization of ZnO NPs

Various analytical techniques were used for the characterization of ZnO NPs in order to determine their size, functional groups, shape, crystallographic structure, and elemental analysis. These techniques include UV-Vis spectroscopy, FTIR, SEM, XRD, and EDX. The optical properties of biosynthesized ZnO NPs were studied using a UV-Vis spectrophotometer (double-beam, Optizen 3320, Mecasys, Daejeon, Republic of Korea) in the wavelength range of 300–700 nm. The PerkinElmer 65 FTIR spectrophotometer was used for the identification of functional groups present in biosynthesized ZnO NPs within the range of 4000–500 cm^−1^. The crystalline structure of biofabricated ZnO NPs was determined using an X-ray diffractometer (Quaid-i-Azam University Islamabad, Pakistan) at 40 kV and 30 mA with CuKα-radiation working between 10° and 80° of 2θ angles at 2°/min. The SEM analysis was performed at the Institute of Space and Technology, Islamabad, Pakistan, for the determination of the morphology and size of biofabricated ZnO NPs operating at 10 kV. The elemental identification of ZnO NPs was determined using the EDX detector fixed to the SEM instrument.

### 4.4. Antibacterial Activity

An antibacterial study of ZnO NPs from algae extract was performed using the agar-well diffusion method against both Gram-positive (*B. pumilus* and *S. aureus*) and Gram-negative (*E. coli* and *P. aeruginosa*) bacteria. As far as antibacterial and antioxidant activities were concerned, a stock solution of 1 g/10 mL of each sample was used. The antimicrobial activity was performed in a sterilized laminar flow. The sterilized nutrient agar medium was poured into petri dishes. The medium was allowed to cool for some time, and the bacterial inoculum was spread gently with the help of cotton bugs over the nutrient agar surface. Then, wells of 6 to 8 mm were made using a sterilized cork borer in the agar medium. The antibiotic rifampicin and sterilized water were used as positive and negative controls, respectively. The 20 µL of algal extract were employed in their respective plates. The zinc oxide NPs were introduced with concentrations of 20 µL, 30 µL, and 40 µL into the wells. The petri plates were cautiously sealed with parafilm. These plates were incubated in an incubator for a day at 37 °C. After a given period, the widths of the growth inhibition zones were scaled in millimeters [16].

### 4.5. Antioxidant Potential

To measure the antioxidant potential, 0.12 mg of DPPH was weighed on a balance and dissolved in 85 mL of pure methanol. The reagent bottle containing DPPH was covered with aluminum foil and kept in a dark place for 1 h. As a standard control, 1 mg of ascorbic acid in 100 mL of distilled water was prepared. Five samples of ZnO NPs with concentrations of 10, 20, 30, 40, and 50 μg/mL were prepared. Then, 800 μg/mL of DPPH was added into each Eppendorf tube of ZnO NPs, and the volume was raised up to 1.5 mL with the addition of methanol in each sample. After that, they were placed in the dark for 1 h. Because of the scavenging activity, a clear color variation from purple/violet to yellow was observed. A UV spectrophotometer was used to measure the absorbance of samples at 517 nm. The antioxidant potential of nanoparticles was compared with that of standard ascorbic acid. The scavenging activities of samples were measured by the following formula:Scavenging Activity (%) = (A control − A sample)/A control × 100

For the determination of the antioxidant potential of algal extract, the same protocol was followed. After the measurement of scavenging activity by taking sample concentrations, the graph was plotted along the x-axis against the percentage inhibition along the y-axis.

## 5. Conclusions

The green fabrication of zinc oxide NPs using algae has received substantial interest due to their rapid growth, environmentally safe nature, and cost-effective protocol. A UV-Vis instrument monitored the color change from light yellow to white, indicated the formation of ZnO NPs. The capping agents found in algal extract have vital roles in the fabrication process, as shown in FT-IR. The proposed important active biomolecules are flavanol glycosides, neo-clerodane, diterpenoids, phytoecdysones, ergosterol, iridoid glycosides, and many other polyphenols. XRD pattern showed the highly crystalline structure of ZnO NPs. SEM images revealed the spherical shape of ZnO NPs. The antioxidant action of the respective nanoparticles was assessed via the DPPH assay, revealed that ZnO NPs are highly antioxidant. Both green-fabricated ZnO NPs, either produced by using zinc nitrate salt precursor or zinc acetate salt precursor, showed higher antibacterial activity against both types of Gram-positive bacteria and Gram-negative bacteria. The higher antibacterial activity was shown against *S. aureus*. The present study intensified the importance of greenly synthesized ZnO NPs in the biomedical field, especially as a potent antioxidant as well as an effective antimicrobial agent.

## Figures and Tables

**Figure 1 micromachines-14-00928-f001:**
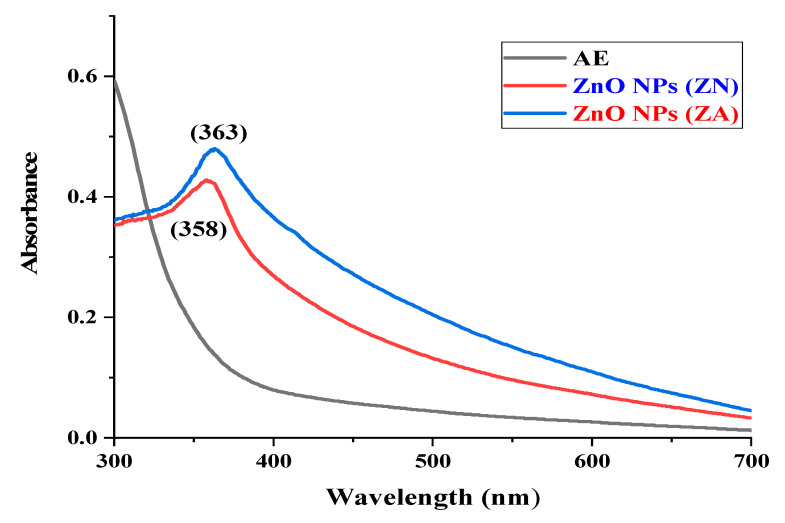
UV-visible absorption spectrum of ZnO NPs from ZA (red) and ZN (blue) precursors and algae extract (black).

**Figure 2 micromachines-14-00928-f002:**
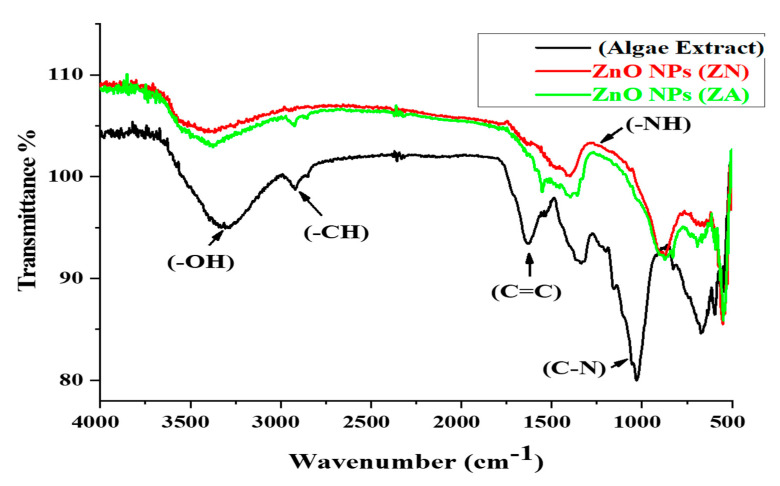
FTIR spectra of algae extract, ZnO NPs (ZA), and ZnO NPs (ZN).

**Figure 3 micromachines-14-00928-f003:**
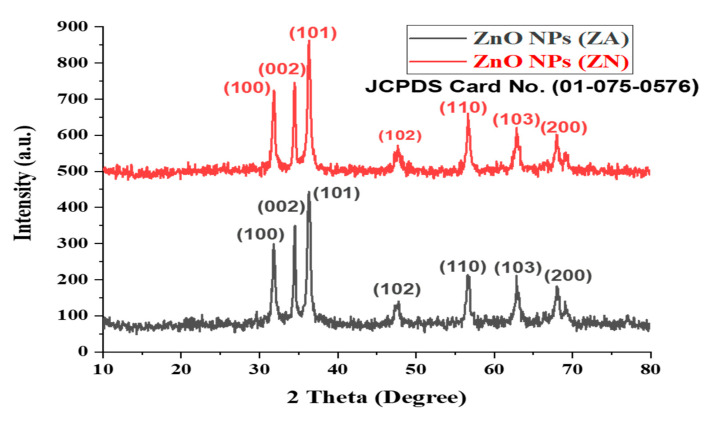
X-ray diffraction pattern of ZnO NPs (ZA) and (ZN) salt precursors.

**Figure 4 micromachines-14-00928-f004:**
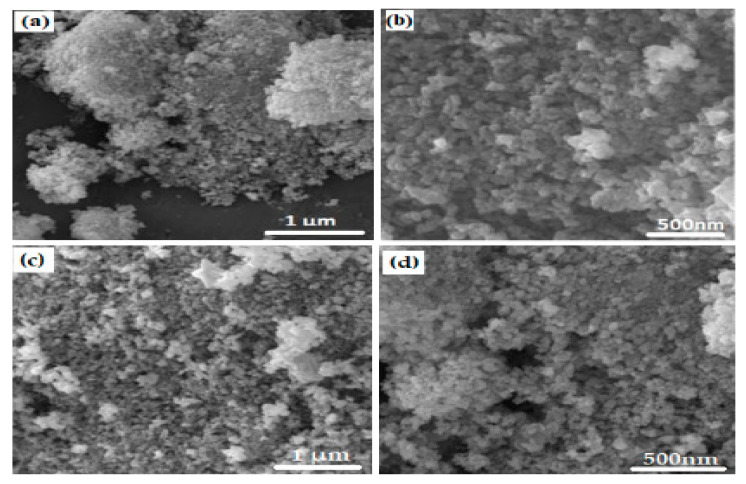
(**a**,**b**) SEM images of ZnO NPs (ZA) at high and low resolutions, (**c**,**d**) SEM images of ZnO NPs (ZN) at high and low resolutions.

**Figure 5 micromachines-14-00928-f005:**
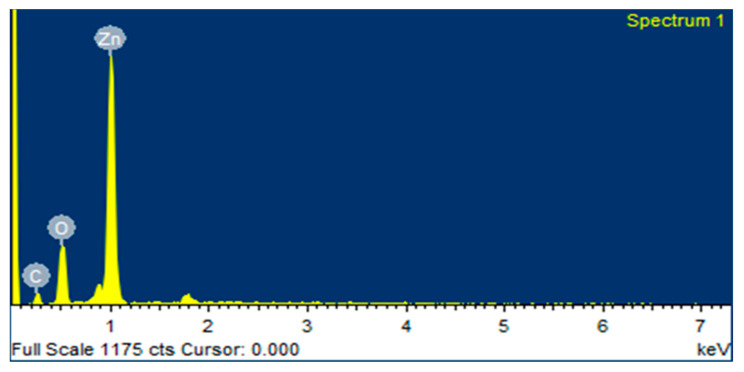
EDX spectrum of ZnO NPs (ZA).

**Figure 6 micromachines-14-00928-f006:**
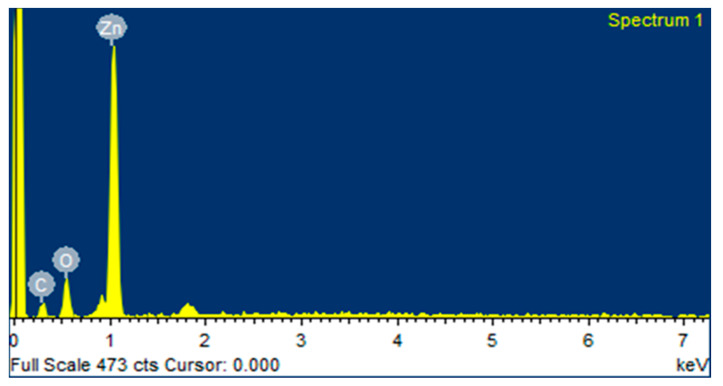
EDX spectrum of ZnO NPs (ZN).

**Figure 7 micromachines-14-00928-f007:**
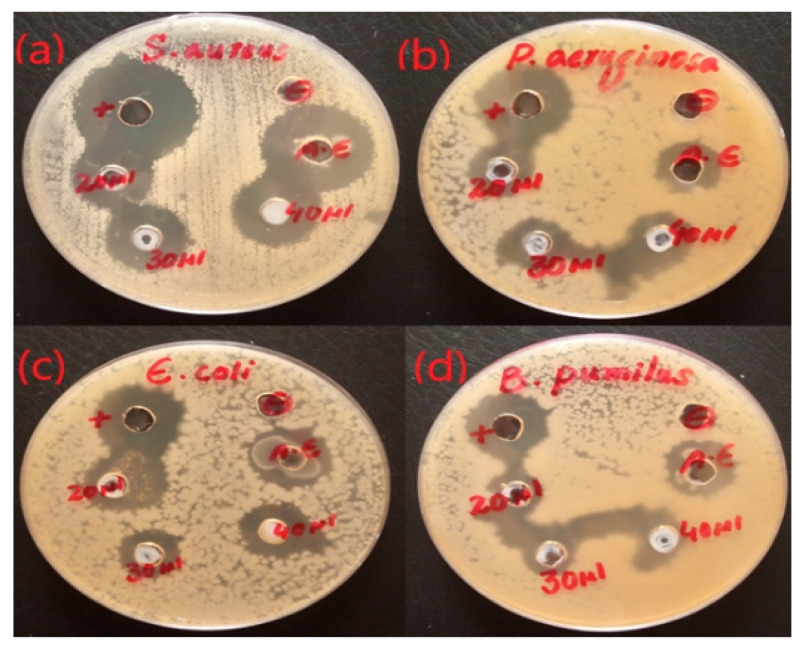
Antibacterial activity of ZnO NPs with zinc acetate salt precursor (ZA).

**Figure 8 micromachines-14-00928-f008:**
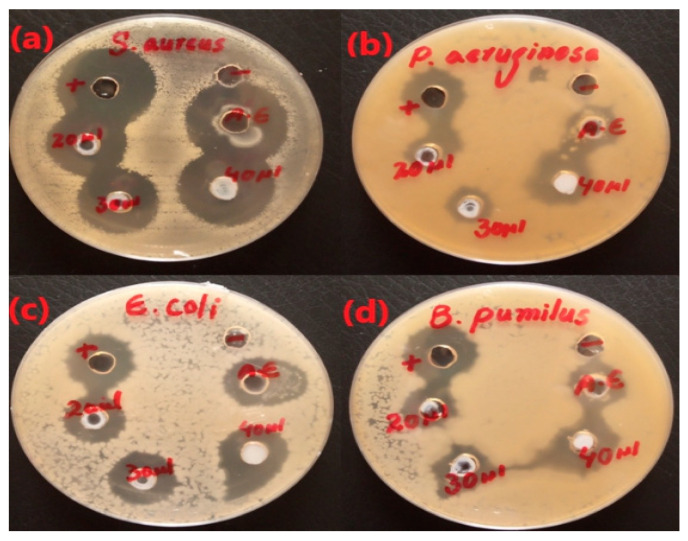
Antibacterial activity of ZnO NPs with zinc nitrate salt precursor (ZN).

**Figure 9 micromachines-14-00928-f009:**
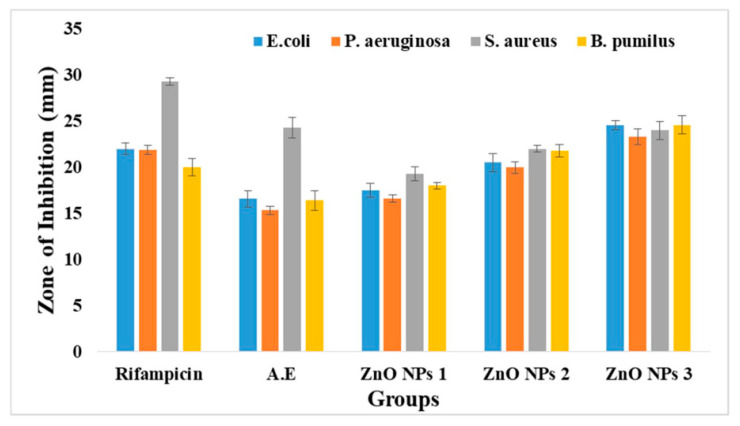
Antibacterial activity graph of ZnO NPs with salt zinc acetate (ZA).

**Figure 10 micromachines-14-00928-f010:**
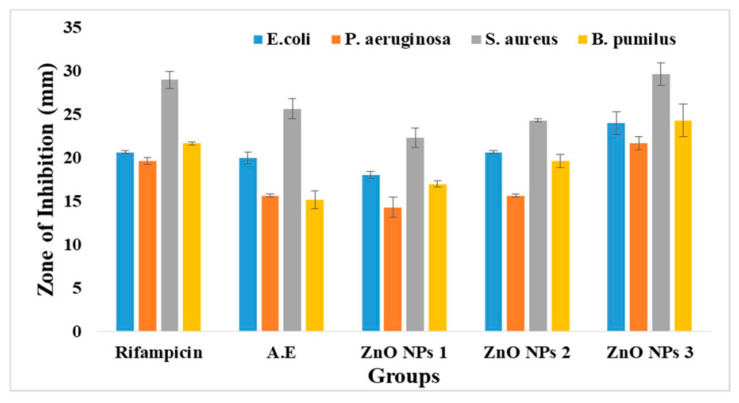
Antibacterial activity graph of ZnO NPs with salt zinc nitrate (ZN).

**Figure 11 micromachines-14-00928-f011:**
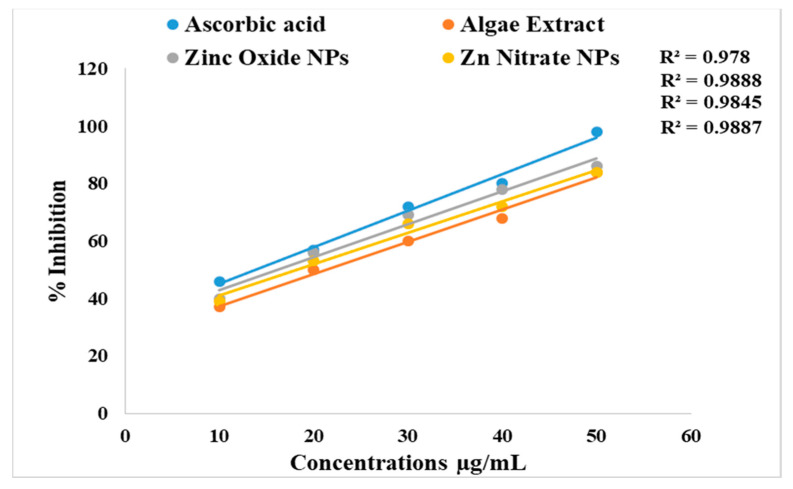
DPPH scavenging activity of ascorbic acid, algae extract, and ZnO NPs (ZA) and (ZN).

**Figure 12 micromachines-14-00928-f012:**
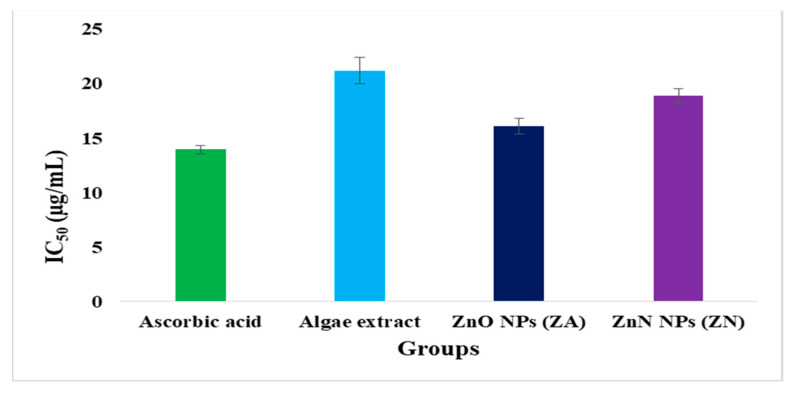
IC_50_ values of standard ascorbic acid, algae extract, and ZnO NPs.

## Data Availability

We declare that the materials described in the manuscript, including all relevant raw data, will be freely available to any scientist wishing to use them for noncommercial purposes, without breaching participant confidentiality.

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
