# Peer review of "Green Synthesis of Zinc Oxide (ZnO) Nanoparticles from Green Algae and Their Assessment in Various Biological Applications"

_micromachines, 2023, doi:10.3390/mi14050928_

Round 1

Reviewer 1 Report

Comments and Suggestions for Authors

I have the following comments for the authors:

1: In the introduction section: this section must be improved, as they have written a very long introduction without much focus. It gives a feel of the introduction of a book which is covering a broad field.

Viz. Page-03, line 160, the characterization techniques mentioned here are standard techniques and they do not require to discuss in this section. Also, the author should omit them from the abstract section.

2: figure-1: is it possible to show this figure in a supplementary information sheet?

3: figure-2: it looks blurry and the UV peak should be marked in the figure itself.

4: figure-3: Please index the FTIR peaks in the figure.

5: figure-6: The SEM image is too elongated to truly understand the microstructure, please insert them correctly, and also the size of the particle is not visible clearly, therefore it may not be appropriate to say the average particle size is 20 nm or so.

6: in the EDX table, the authors have given the wt% of Carbon also, they may consider removing it.

Over all work presented in the manuscript is not very focused and It must be improved.

Thanks.

Author Response

Response to Reviewer 1

Introduction is written again with relevant references.

All references are checked, and irrelevant references are deleted.

Research design, methods, results, and conclusions are improved as per reviewer’s suggestion.

Response:

Thank you for the comment, we have improved the English.

I have the following comments for the authors:

1: In the introduction section: this section must be improved, as they have written a very long introduction without much focus. It gives a feel of the introduction of a book which is covering a broad field.

Response:

Thank you for the comment, we have shortened introduction with reference to the title of the manuscript.

Viz. Page-03, line 160, the characterization techniques mentioned here are standard techniques and they do not require to discuss in this section. Also, the author should omit them from the abstract section.

Response:

Thank you for the comment, done as suggested.

2: figure-1: is it possible to show this figure in a supplementary information sheet?

Response:

Thank you for the comment, there is no need of this figure. Therefore, it’s deleted.

3: figure-2: it looks blurry and the UV peak should be marked in the figure itself.

Response:

Thank you for the comment, figure 2 is changed with clear figure and peak mark.

4: figure-3: Please index the FTIR peaks in the figure.

Response:

Thank you for the comment, peaks are indexed in Figure 3.

5: figure-6: The SEM image is too elongated to truly understand the microstructure, please insert them correctly, and also the size of the particle is not visible clearly, therefore it may not be appropriate to say the average particle size is 20 nm or so.

Response:

Thank you for the comment, SEM images are replaced.

6: in the EDX table, the authors have given the wt% of Carbon also, they may consider removing it.

Response:

Thank you for the comment, the Carbon wt% is due to puttering step.

Overall work presented in the manuscript is not very focused and It must be improved.

Response:

Overall, all the manuscript is improved.

Thanks.

Reviewer 2 Report

Comments and Suggestions for Authors

Author Response

Response to reviewer 2

Introduction is written again with relevant references.

All references are checked, and irrelevant references are deleted.

Research design, methods, results, and conclusions are improved as per reviewer’s suggestion.

Response:

Thanks for comment, we have improved the English.

Overall, the manuscript is good and novelty is lacking in this manuscript.

Response:

Thanks for comment, the novelty is added in introduction now.

Some comments are given below which can improve the manuscript.

  • Authors can add details of antibacterial activity with related to colony count method
  • Provide high resolution images.
  • Introduction is lengthy, need to limit it
  • Grammatical/typological error need to addressed throughout the manuscript 
  • English need to be revised

Response:

Thanks for comment, we have improved the manuscript in terms of all above suggestions.

Reviewer 3 Report

This review report has been removed from the review record as it did not meet MDPI’s review report standards (https://www.mdpi.com/reviewers#_bookmark11).

Reviewer 4 Report

Comments and Suggestions for Authors

Dear Authors, I found the article fluid to read and stimulating from a scientific point of view, below are my observations. I hope they help to enhance the quality of this work.

Chapter 1 

Among the techniques mentioned, dynamic light scattering (DSC) is not even mentioned. This technique is fundamental for the characterization of the size and distribution of nanoparticles synthesized in solution.

Line 83 - “One” should not be capitalized

Line 112 - The cited reference (15) does not explain the sentence mentioned above (“These ordinary strains and Algae extract secrete phytochemicals that act as both reducing agent and capping agent or stabilizing agent”). Please put an adequate and clearer reference.

Line 122 - Here we are talking about the role of algae extract in nanoparticle synthesis. This reference is inadequate compared to the previous sentence.  Could you please explain better?

Lines 118-119 - I share your thoughts and I am very happy to have read this concept in a scientific article. Thank you for the desire to convey this thought to everyone who will read this article.

Line 149 - Describing nanoparticle sizes as "tiny" is not appropriate, especially as the next sentence states that the typical particle size used in this work is 100 nm and other types of nanoparticles exist that have sizes on the order of 1-5 nm.

Section 2.2
Please, can you explain better the role of Algal extract in the nanoparticles synthesis process from the chemical point of view? Does it act as a reducing, capping agent or both?

Can you explain how this helps make the synthesis greener? Furthermore, the proposed synthesis is a green synthesis, not a biosynthesis.

Line 264 - To say it's the best method is a bit too bold.

Lines 280 - 282 The sentence “Prepare 0.05 M Zn (NO3) in 2.6 H2O for the synthesis of nanoparticles from zinc nitrate hexahydrate. Add 10 ml of algal extract to 50 ml of salt solution, and the same protocol will be followed as mentioned above” it is not amalgamated to the chapter and it is not clear.

Section 2.4 and 2.5 

- The concentration is indicated in "l", a letter which is commonly used to indicate "liters" which is a unit of volume. Please correct the units of measurement

Section 3.1 

- The AE curve does not have a peak at 302 nm, but an absorption band.

Section 3.2
- Please better specify the peak assignment in the 3000-3050 cm-1 band, since in that range it is not present an absorption peak. Also why hasn't the peak been assigned to about 2800 cm-1? Is it possible to get information about it? 

-Also, please write the unit cm-1 correctly.

Section 3.3
- SEM images are distorted, small and difficult to understand. Please replace them with clearer and high resolution images. 

-In this regard,an additional characterization of the nanoparticle size distribution by Dynamic Light Scattering (DLS) could be useful for the purposes of the work.

Chapter 3 

- The following items are not referred to in the text: Figure 3, Figure 4, Figure 5, Figure 6, Figure 7, Figure 8, Figure 9, Figure 10, Figure 11, Figure 12, Figure 13, Figure 14, Figure 15, Figure 17 and Table 1

Line 442 - In the description of table 2 it says "Table 1".

Chapter 4 

- This section, in my opinion, is too much like an introduction. I suggest to highlight the obtained results.

Lines 586-598 - those lines are repeated in lines 599-611.

Author Response

Response to reviewer

Introduction is written again with relevant references.

All references are checked, and irrelevant references are deleted.

Research design, methods, results, and conclusions are improved as per reviewer’s suggestion.

Response:

Thanks for comment, English in terms of spelling is improved.

Dear Authors, I found the article fluid to read and stimulating from a scientific point of view, below are my observations. I hope they help to enhance the quality of this work.

 Chapter 1 

Among the techniques mentioned, dynamic light scattering (DLS) is not even mentioned. This technique is fundamental for the characterization of the size and distribution of nanoparticles synthesized in solution.

Response:

Thanks for comment, the DSC is not done in this study, that is why it is missing in manuscript.

Line 83 - “One” should not be capitalized

Response:

Thanks for comment, it is corrected.

Line 112 - The cited reference (15) does not explain the sentence mentioned above (“These ordinary strains and Algae extract secrete phytochemicals that act as both reducing agent and capping agent or stabilizing agent”). Please put an adequate and clearer reference.

 Response:

Thanks for the comment, clear reference is added.

Line 122 - Here we are talking about the role of algae extract in nanoparticle synthesis. This reference is inadequate compared to the previous sentence.  Could you please explain better?

Response:

Thanks for the comment, we have explained in manuscript.

Lines 118-119 - I share your thoughts and I am very happy to have read this concept in a scientific article. Thank you for the desire to convey this thought to everyone who will read this article.

Response:

Thanks for your suggestions.

Line 149 - Describing nanoparticle sizes as "tiny" is not appropriate, especially as the next sentence states that the typical particle size used in this work is 100 nm and other types of nanoparticles exist that have sizes on the order of 1-5 nm.

Response:

Thanks for your suggestions. We have changed it

Section 2.2
Please, can you explain better the role of Algal extract in the nanoparticles synthesis process from the chemical point of view? Does it act as a reducing, capping agent or both?

Can you explain how this helps make the synthesis greener? Furthermore, the proposed synthesis is a green synthesis, not a biosynthesis.

Response:

Thanks for the comment, algal extract play both role. It is green synthesis. We have changed it.

Line 264 - To say it's the best method is a bit too bold.

Response:

Thanks for your suggestions. We have removed this line.

Lines 280 - 282 The sentence “Prepare 0.05 M Zn (NO3) in 2.6 H2O for the synthesis of nanoparticles from zinc nitrate hexahydrate. Add 10 ml of algal extract to 50 ml of salt solution, and the same protocol will be followed as mentioned above” it is not amalgamated to the chapter and it is not clear.

Response:

Thanks for comments, we have changed the methodology. It is corrected now.

Section 2.4 and 2.5 

The concentration is indicated in "l", a letter which is commonly used to indicate "liters" which is a unit of volume. Please correct the units of measurement

Response:

Thanks for comments, it is corrected.

Section 3.1 

The AE curve does not have a peak at 302 nm, but an absorption band.

Response:

Thanks for comments, it is corrected.

Section 3.2
Please better specify the peak assignment in the 3000-3050 cm-1 band, since in that range it is not present an absorption peak. Also why hasn't the peak been assigned to about 2800 cm-1? Is it possible to get information about it? 

Also, please write the unit cm-1 correctly.

Response:

Thanks for comments, it is corrected at 2800 cm-1, format is also changed from cm-1 to cm-1

Section 3.3
SEM images are distorted, small and difficult to understand. Please replace them with clearer and high resolution images. 

In this regard, an additional characterization of the nanoparticle size distribution by Dynamic Light Scattering (DLS) could be useful for the purposes of the work.

Response:

Corrected.

Thanks for comments, we have changed SEM images and DSC is not done in this study, that is why it is missing in manuscript.

Chapter 3 

The following items are not referred to in the text: Figure 3, Figure 4, Figure 5, Figure 6, Figure 7, Figure 8, Figure 9, Figure 10, Figure 11, Figure 12, Figure 13, Figure 14, Figure 15, Figure 17 and Table 1

Response:

Thanks for comments, we have done it.

Line 442 - In the description of table 2 it says "Table 1".

Response:

Thanks for comments, we have done it.

Chapter 4 

This section, in my opinion, is too much like an introduction. I suggest to highlight the obtained results.

Thanks for comments, we have done it.

Done.

Lines 586-598 - those lines are repeated in lines 599-611.

Thanks for comments, we have deleted it.

Round 2

Reviewer 1 Report

Comments and Suggestions for Authors

I have the following comments for the authors:

1: Please index the XRD peaks in Figure-03.

2: FESEM image does not have a very good resolution to see nano-particles upto 20 nm or so, therefore it may not be appropriate to say the average particle size is 20 nm or so. Authors may reconsider this.

3: Please improve the quality of figure-06, the written words are not visible.

Thanks.

Author Response

Reviewer 1

I have the following comments for the authors:

Does the introduction provide sufficient background and include all relevant references?

Improved

Are all the cited references relevant to the research?

Improved

Is the research design appropriate?

Improved

Are the methods adequately described?

Improved

Are the results clearly presented?

Improved

Are the conclusions supported by the results?

Improved

1: Please index the XRD peaks in Figure-03.

Response: Thank You, XRD graph peaks are indexed.

2: FESEM image does not have a very good resolution to see nano-particles upto 20 nm or so, therefore it may not be appropriate to say the average particle size is 20 nm or so. Authors may reconsider this.

Response: Thank you for your suggestion, we improve the resolution of SEM Image, and We can evaluate the SEM image size by using ImageJ and origin software. Average particles size is 20nm.

3: Please improve the quality of figure-06, the written words are not visible.

Response: Thank you, it is changed.

Reviewer 2 Report

Comments and Suggestions for Authors

Can be considered 

Author Response

Reviewer 3:

Response: Thank you for valuable suggestion, we improve the material method.

Does the introduction provide sufficient background and include all relevant references?

Are all the cited references relevant to the research?

Is the research design appropriate?

Are the methods adequately described?

Improved

Are the results clearly presented?

Are the conclusions supported by the results?

Reviewer 4 Report

Comments and Suggestions for Authors

Dear authors, thank you for the additions, however I have noticed some very important shortcomings in the work.

I also suggest an intensive review regarding typos, characters repeated by mistake, merged words and letters present in the text without context.

Line 181: figure 3.1  nm It does not make sense

Line 191:  There is no Fig 3.2. Here we are referring to the FTIR spectrum which is in Figure 2

Line 215: There is no figure 3.3, you probably refer to Figure 3

Line 235: There is no figure 3.4, you probably refer to Figure 4

Section 3.4: It is not possible to estimate the size of the NPs via the SEM images shown. Please provide information on how the presented values were obtained.
I suggest implementing dynamic light scattering (DLS) measurements.

Line 246-252: Eliminate

Line 265: The numbering to follow for this figure is Figure 5 (fix in line 225)

Line 270: There is no fig 3.6, you probably refer to Figure 6

Line 286: There is no fig 3.7, you probably refer to Figure 7

Line 297: remove this former legend

Author Response

Reviewer 4

Dear authors, thank you for the additions, however I have noticed some very important shortcomings in the work.

I also suggest an intensive review regarding typos, characters repeated by mistake, merged words and letters present in the text without context.

Does the introduction provide sufficient background and include all relevant references?

Improved

Are all the cited references relevant to the research?

Improved

Is the research design appropriate?

Improved

Are the methods adequately described?

Improved

Are the results clearly presented?

Improved

Are the conclusions supported by the results?

Improved

Edits/Scientific Improvements:
1. Line 181: figure 3.1  nm It does not make sense

Response: Thank you, it is changed with suitable sentence.

2.Line 191:  There is no Fig 3.2. Here we are referring to the FTIR spectrum which is in Figure 2

Response: Thanks, correct the Figure number 3.2.
3.Line 215: There is no figure 3.3, you probably refer to Figure 3
Response: It is changed, Thank you.

Line 235: There is no figure 3.4, you probably refer to Figure 4

Response: It is changed, thank you.
Section 3.4: It is not possible to estimate the size of the NPs via the SEM images shown. Please provide information on how the presented values were obtained. I suggest implementing dynamic light scattering (DLS) measurements.

Response: Because DLS is not done in this study, in this study used Image. J software for the size evaluation of nanoparticles.

Line 246-252: Eliminate
Response: Thanks for your suggestion, 246-252 lines are eliminated.

Line 265: The numbering to follow for this figure is Figure 5 (fix in line 225)
Response: It is done.

Line 270: There is no fig 3.6, you probably refer to Figure 6
Response: It is Changed, Thank you.

Line 286: There is no fig 3.7, you probably refer to Figure 7

Response: It is changed, Thanks.
Line 297: remove this former legend

Response: It is removed.

Round 3

Reviewer 4 Report

Comments and Suggestions for Authors

Dear authors, thank you for the additions, in this form the work is much more accurate, clear and exhaustive.

However, I have one last note to make.

In section 2.4 authors provides measurements of NPs with an accuracy that cannot be obtained from the images shown. It could also be useful to indicate the pixel dimension of the image used to acquire the measurements, in fact, seeing the scale, the uncertainty in the measurement is probably of the same order of magnitude as the measurements themselves. For this reason I think it is more appropriate to explain that the approximate dimensions of the NPs obtained from SEM analysis are confirmed by other measures such as optical absorption.

Author Response

SEM is used to characterize, visualize surface morphology, particle size distribution, particle shape and agglomeration of nanoparticles. In current paper, section 2.4 (figure 4: a-d) the SEM images of ZnO NPs are taken as low and high resolutions of 500 nm and 1µm, as these (500 nm & 1µm) are the resolution values in the SEM analyzer but not the size of ZnO NPs. Although there are also other measurements for particle size measurement, in current study we found the particle size by SEM analysis (Image J and Origin software were used to determine the particle size). Image J software was used to analyze particles by setting the minimum size and maximum pixel area size and selecting the average 100 NPs. The origin software was used to make size distribution histogram with the data found by Image J to indicate the average size of ZnO NPs (ZN) ranges from 50 nm to 80 nm, with an average size of 65 nm, and ZnO (ZA) size ranges from 20 nm to 60 nm, with an average size of 40 nm.
